# Detecting Desert Locust Breeding Grounds: A Satellite-Assisted Modeling Approach

W. Lee Ellenburg [1,2,*], Vikalp Mishra [1,2], Jason B. Roberts [2], Ashutosh S. Limaye [2], Jonathan L. Case [3,4], Clay B. Blankenship [4,5] and Keith Cressman [6]

1   Earth System Science Center, The University of Alabama in Huntsville, 320 Sparkman Dr., Huntsville, AL 35805, USA; vikalp.mishra@nasa.gov
2   NASA SERVIR Science Coordination Office, Marshall Space Flight Center, 320 Sparkman Dr., Huntsville, AL 35805, USA; jason.b.roberts@nasa.gov (J.B.R.); ashutosh.limaye@nasa.gov (A.S.L.)
3   ENSCO, Inc., Huntsville, AL 35805, USA; Jonathan.Case-1@nasa.gov
4   NASA Short-Term Prediction Research and Transition (SPoRT) Center, Marshall Space Flight Center, Huntsville, AL 35805, USA; clay.blankenship@nasa.gov
5   Universities Space Research Association, Huntsville, AL 35805, USA
6   United Nations Food and Agriculture Organization, 00153 Rome, Italy; Keith.Cressman@fao.org
*   Correspondence: lee.ellenburg@uah.edu

**Abstract:** The objective of this study is to evaluate the ability of soil physical characteristics (i.e., texture and moisture conditions) to better understand the breeding conditions of desert locust (DL). Though soil moisture and texture are well-known and necessary environmental conditions for DL breeding, in this study, we highlight the ability of model-derived soil moisture estimates to contribute towards broader desert locust monitoring activities. We focus on the recent DL upsurge in East Africa from October 2019 though June 2020, utilizing known locust observations from the United Nations Food and Agriculture Organization (FAO). We compare this information to results from the current literature and combine the two datasets to create "optimal thresholds" of breeding conditions. When considering the most optimal conditions (all thresholds met), the soil texture combined with modeled soil moisture content predicted the estimated DL egg-laying period 62.5% of the time. Accounting for the data errors and uncertainties, a $3 \times 3$ pixel buffer increased this to 85.2%. By including soil moisture, the areas of optimal egg laying conditions decreased from 33% to less than 20% on average.

**Keywords:** desert locust; soil moisture; modeling

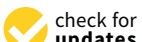



## 1. Introduction

The recent (2019–2020) upsurge of desert locust (*Schistocerca gregaria*) in East Africa was the worst the region has seen over 40 years. Countries such as Kenya saw their worst upsurge in 70 years. The swarms of desert locust (DL) can be dense and highly mobile, devouring the vegetation that they come across; the result can mean complete devastation of crops and grasslands [1]. DL control measures, such as pesticides and mechanical removal, are most effective when the locusts are immature and wingless, limiting their mobility [2]. Understanding what conditions are optimal for DL breeding is an important component in locating the wingless "hoppers". Soil conditions, such as texture, moisture, and temperature, are the most limiting factors for the success of DL breeding and egg incubation [3–7]. There have been several recent studies in the literature highlighting the importance of these soil conditions using satellite-derived estimates (e.g., [8–12]). In this study, we aim to evaluate the utility of an existing model-derived soil moisture product in the region to complement the ongoing DL monitoring efforts. While soil moisture (SM) is a known limiting factor and can be estimated using satellite remote sensing, the use of model-derived SM has the ability to estimate the variable at high spatial (vertical) and temporal resolutions and lowers the barrier to operationalization and forecasting.

Expert knowledge and advanced numerical models are used to understand the trajectory of existing swarms [13] and to assess and link between environmental conditions and a species (e.g., [3,14]). However, DL monitoring efforts look at current (near-real-time) conditions for habitat suitability, focusing on immature (wingless) hoppers, where control measures are more effective [2]. For monitoring potential habitats, the Food and Agriculture Organization (FAO) currently uses and advises countries to use vegetation greenness and rainfall information. While vegetation is an important indicator of DL habitat, datasets that can estimate or infer SM add an important temporal component to monitoring and forecasting efforts. SM can be an early signal for vegetation growth, and it is a limiting factor in the egg-laying activity of DL. Field and laboratory studies show that female DL prefer sandy substrates with moisture between 5% and 25% at a depth between 2 and 15 cm [15–17].

The FAO currently uses satellite-derived rainfall estimates at a resolution of 25 km [18,19], and these are used to deduce the subsequent increase in SM. However, there are multiple datasets available from satellite sensors or models that estimate SM directly at relatively higher spatial resolutions. Though satellite-based datasets have the potential to cover vast inaccessible areas, they are typically available at coarser spatial resolutions than those required by most applications. Piou et al. [11] used a disaggregation approach to downscale the original 40 km Soil Moisture and Ocean Salinity (SMOS) SM to 1 km. They found that moisture conditions 70 to 90 days prior were best at predicting the presence of DL at an optimal SM of 0.09 $cm^3/cm^3$ and posited that gains in early warning timing of up to 3 weeks were achieved as compared to vegetation imagery alone.

While disaggregation approaches have shown promise (e.g., [10,11,20]), the process can be challenging, particularly due to its reliance on additional high-resolution data resources, complicating efforts to operationalize in existing monitoring efforts. Additionally, microwave sensors typically sense the SM in the top 3–5 cm of the soil layer [21–24]. DL egg laying, survival, and hatching depend upon the moisture conditions between depths of 5 and 15 cm from the surface. Hence, satellite-driven SM estimations alone may not be sufficient for identifying the optimal DL breeding conditions.

Land surface models (LSMs) have their own biases and limitations (i.e., parameterization, model complexities and physics, etc.); however, such models provide moisture condition information at a much larger range of soil layer depths (0–200 cm) at relatively higher spatial (1–10 km) and temporal resolutions. Additionally, standard configurations for widely used LSMs, such as the NASA's Land Information Systems [25,26], are often used in operational settings, and they provide forecasting capabilities and the potential for satellite data assimilation. Though recent studies on the importance of SM in DL prediction have begun to consider model-based approaches to expand the depth of moisture estimations, to our knowledge, there has not been an evaluation of an LSM-derived SM product for this purpose. Gómez et al. [8] utilized the Soil Moisture Active Passive (SMAP) level-four root zone estimation; however, the root zone layer used in that product combined depths down to 100 cm, well beyond the optimal depths for DL egg laying.

We hypothesize that use of high-resolution (3 km) modeled SM datasets from an LSM can accurately represent conditions beyond the surface layer to DL breeding depths. Furthermore, due to the low barrier for operationalization, the modeling systems have the potential to develop or improve information products all along the locust management decision-making chain, providing forecasted SM conditions that could aid in preparedness and control measures. While recognizing the many additional environmental conditions required to support the locust life cycle, the goal of this study is to simply highlight the ability of model-derived SM estimates to contribute towards broader locust monitoring activities.

## 2. Data and Methods

SERVIR (servirglobal.net (accessed on 20 March 2021)), in collaboration with NASA's Short-Term Prediction Research and Transition Center (SPoRT), has an instance of NASA's Land Information Systems (LISs; [25,26]) currently operational over Eastern and Southern

Africa at a 3 km spatial resolution using near-real-time vegetation information. Together with the high-resolution (250 m) International Soil Reference and Information Centre (ISRIC) SoilGrids database [27], we developed a thresholding approach in order to detect the likelihood of DL breeding potential. To identify the optimal breeding conditions, we used known locust observations from the FAO, looked back to the estimated breeding period, and considered the soil conditions that were present. First, the 10th and 90th percentile values of soil texture (clay and sand) were determined, and the percent area coverage was evaluated for each. Next, the DL development cycle was used to determine an approximate egg-laying time frame. Soil moisture was aggregated to the weekly time scale to determine the estimated conditions during egg laying, and the ranges of values and area of coverage were evaluated and compared against those from the current literature. The two attributes were then combined to create a range of breeding condition thresholds and evaluated against the known DL observations. To better understand the temporal and spatial uncertainty, a 3 × 3 (9 × 9 km) buffer was applied to the final threshold and discussed. With the lack of "no locust" observations, the results were evaluated in terms of classified areas and discussed in the context of current monitoring efforts and existing literature.

### 2.1. Study Area and Time Period

The area selected for this study is in Eastern Africa (Figure 1). The countries of Eritrea, Djibouti, Somalia, Ethiopia, and Kenya were particularly in focus. The climate of East Africa ranges from hot arid to tropical savanna [28] as a result of the complex terrain; in general, it is anomalously dry for equatorial regions [29]. Rainfall across the region can be characterized by a bimodal regime, with the movement of the Intertropical Convergence Zone resulting in the "long rains" from March to May and the "short rains" from October to December [29]. The study period was from December 2019 through June 2020, capturing the start and height of the locust upsurge in the region.

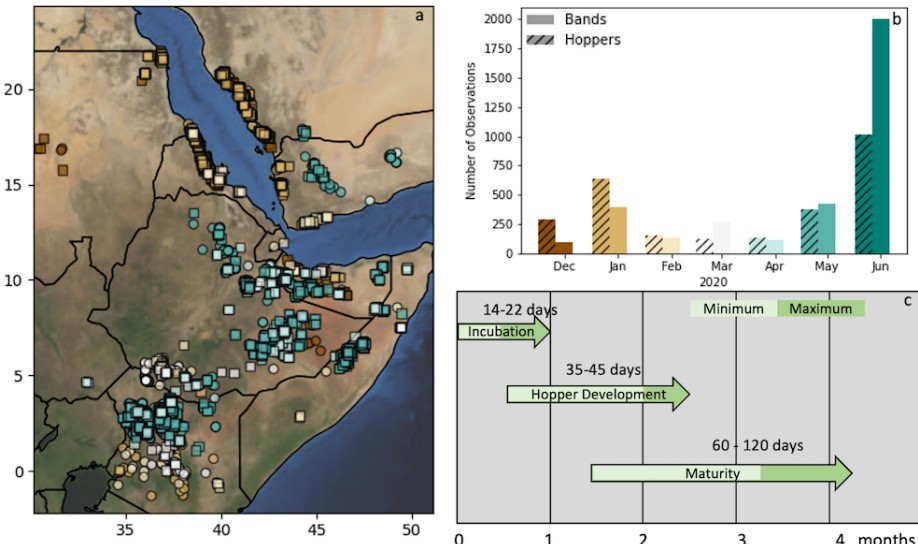

**Figure 1.** Study Area (WGS84) (**a**) depicting the locust observations in East Africa from December 2019 through June 2020. Solitary hoppers are noted by a circle and bands by squares. The colors of the observations relate to the timing in the (**b**) bar plot of observation counts by month. (**c**) A depiction of the timeline of the desert locust (DL) life cycle from breeding to maturity in the East African environment, as reported in WMO/FAO [30].

### 2.2. Locust Observations

The locust observation data were provided by the FAO through their LocustHub platform (https://locust-hub-hqfao.hub.arcgis.com/ (accessed on 20 March 2021)). The

dataset contains information such as the locust life cycle stages and gregariousness—immature and mature winged adults that are either swarming or not, as well as nymphal wingless "hoppers" that are either solitary or gregarious "bands" that march across the land. For this study, we only used the Hopper and Band datasets. By focusing on immature hoppers (solitary and banded), we can assume that the hatching site must be within "marching" distance from their observed location (which can range from hundreds of meters to several kilometers [31]) and can therefore be used as a safe proxy for egg-laying locations (particularly true given that the resolution of the SM data is roughly 3 km).

In total, 6171 observations were used—2734 hoppers and 3437 bands. The timing of the observations was similar between the two groups, with two significant population buildups seen in January and May/June (Figure 1). These increases are roughly in line with the breeding cycles of DL observed from the FAO Locust analysis [32]. It can be seen from Figure 1 that band/hopper observations in late 2019/early 2020 (brown) exist mostly in the north and east (in the countries including Eritrea, Djibouti, and Yemen). These matured into swarms that moved southwards, where favorable conditions facilitated a second round of breeding that resulted in increased hopper/band observations in May and June (teal/green).

In general, the DL incubation and development periods are dependent on soil and air temperature (WMO/FAO [30], Figure 1c). The incubation period can range between 14 and 22 days in the soil temperatures of the region: approximately 27–32 °C. Immature DL will then remain grounded through their various development stages (instars) for another 35–45 days. Thus, depending on the stage of the hopper, the optimum environmental conditions for egg laying would have to be met approximately 3–10 weeks prior to the observation date. The timing of instar development can vary widely depending on soil and air temperature. The majority (>80%) of the reported observations were in their later instars—three or greater. However, most observations noted several stages, and many were missing altogether. For this analysis, we assumed that the hoppers observed were usually in the later instars, pushing the range to more like 8–10 weeks (in line with the previous studies referenced above).

*2.3. Soil Moisture and Texture Datasets*

2.3.1. Soil Moisture from LIS

The daily averaged SM was modeled using the Noah land surface model [33] within the NASA Land Information System [25], which was set up to run in near real time over East Africa by SERVIR and NASA's SPoRT [34]. The full domain was a 0.03 degree (~3 km) lat/long grid spanning 21° E to 53° E and 22° S to 22° N. The model was forced using precipitation data from the Integrated Multi-satellite Retrievals for Global Precipitation Measurement mission (IMERG) whereas the surface temperature, pressure, winds, and downwelling long/shortwave radiation data were available from three-hourly Global Data Assimilation System analyses and forecasts [35,36]. Static fields, such as soil type classification, were from the global FAO soil map [37] using the State Soil Geographic (STATSGO; [38]) categories. The land cover was aggregated from the 500 m MODIS Land Cover Type (MCD12Q1) product [39] using the 20-category National Center for Environmental Prediction (NCEP)-modified International Geosphere-Biosphere Programme (IGBP) land-use classification. The soil bottom temperature climatology (lower boundary condition) was based on 6 years of NCEP Global Data Assimilation System (GDAS) 2 m air temperatures using the method of Chen and Dudhia [40].

The LIS runs also used a daily real-time satellite-derived Greenness Vegetation Fraction (GVF) from Visible Infrared Imaging Radiometer Suite (VIIRS) instrument [41]. Using the satellite-observed GVF improves the partitioning of latent and sensible heating compared to a seasonal climatology [26]. The SM product available from the LIS was the hourly mean moisture content at layer depths of of 0–10, 10–40, 40–100, and 100–200 cm. In this study, the daily composite of the volumetric SM from the top layer (0–10 cm) was used, since the DL is most likely to lay eggs in the 5–15 cm depth range.

### 2.3.2. ISRIC Soil Texture

In addition to the FAO soil texture data used in LIS, we utilized the International Soil Reference and Information Centre (ISRIC) SoilGrids database [27]. SoilGrids is a global 250 m database of standard numeric soil property predictions that include soil texture at depths from the surface to 200 cm. On average, the dataset explains 79% and 73% of the variance in sand and clay percentages (by weight), respectively. In this study, we aggregated the ISRIC dataset to 3 km (nearest neighbor) to match the LIS soil moisture output. At the 3 km level specification of the model, there is not much difference between the two datasets (FAO and ISRIC); however, when used alone, the ISRIC is much preferred due to the spatial resolution and the specificity of texture. Therefore, we used the ISRIC data for texture analysis in this study.

## 3. Results and Discussion

Both the soil texture and SM datasets were used to evaluate their effectiveness in predicting conditions suitable for locust breeding sites. The solitary hoppers and gregarious bands were analyzed separately, as they were disaggregated in the original FAO data. However, since no significant differences were found between the datasets, from here on, we consider them as one and present the combined results for a more cohesive discussion.

### 3.1. Soil Texture Analysis

Female DL require sandy soils to lay their eggs. The ISRIC soil texture data were evaluated at a depth of 5–15 cm for both sand and clay contents. Knowing that the hoppers can move up to several kilometers across the land surface (Ariel and Ayali, 2015), the ISRIC data were aggregated (nearest neighbor) up to the 3 km scale for this historical analysis, matching that of the LIS grid (it should be noted, however, that in operational DL monitoring, the native resolution of 250 m will be extremely useful). The mean percentages (by weight) of sand and clay at DL observations were 58% and 21%, respectively. This aligns well with the assumption of a sandy soil or substrate, where sand percentages <50% are classified as sandy according to the USDA soil classification triangle. The 10/90th percentiles for sand exist between 45 and 69.5%, while the clay percentiles are a bit narrower at 13.5–29.5%.

The sand threshold was met in 80% of the DL observations and covered more than 44% of the study area. This exemplifies the importance of including a secondary texture class. The percentage of area in which the DL clay threshold was met covered only 36% of the area and added further information for refining the "optimal" soil type conditions for breeding. The combined categories reduced the optimal area to 30% and still comprised over 74% of the DL observations on average.

### 3.2. Soil Moisture Analysis

DL also require a specific moisture content for their eggs to remain viable. The averaged weekly volumetric SM content was evaluated to account for daily fluctuations and provide a clearer picture of the SM trends. The 10th and 90th percentiles of weekly averages were analyzed over the (assumed) incubation period, 9 weeks prior. The weekly average of the optimal SM 9 weeks prior was 0.19 $cm^3/cm^3$, with a range encompassing 80% of the data of between 0.12 and 0.27 $cm^3/cm^3$. The analysis was initially conducted for weekly averages over the broader potential incubation periods of 6–10 weeks. The difference in median SM values across the range was 0.04 $cm^3/cm^3$ (~20%), with the earlier (5–7 week) median towards the upper limit of values reported in the literature.

The nine-week averaged SM aligns well with the optimal SM conditions for locust breeding reported within the field- and lab-based literature [15,16] and is consistent with previous studies, such as those of Piou et al. [11] and Gómez et al. [8], which found that a mean of 9 weeks was most indicative of the most optimal breeding conditions. However, the studies using remote-sensing-derived SM reported median surface values of 0.09–0.1 $cm^3/cm^3$. The discrepancy between previous studies using remotely sensed

observations could be due to the fact that the LIS data were reporting the top layer to a depth of 10 cm, while satellite-derived SM typically senses the top-most layer of soil. As the top layer of soil is more susceptible to drying, especially in sandy soils, it makes sense that that modeled product would have a higher—possibly more representative—average SM for the egg-laying depth.

A bootstrap resampling was performed (*n* = 100), and it was found that the SM distribution was quite stable, with a variance in both the 10th and 90th percentile values of less than 1% (*p*-value ≪ 0.05), giving confidence in the distribution. The difference in the mean SM between the two main breeding cycles (October/November and March/April) was ∼0.02, with the later season exhibiting a slightly wider range (mean = 0.19, range = 0.14) than the earlier (mean = 0.19, range = 0.10). However, the locusts observed between the major breeding cycles (December/January/February), though representing <15% of the data, showed a distinctly larger range of values (mean = 0.20, range = 0.19). On average, the areas within the SM threshold constitute 50.1% of the study area.

### 3.3. Combined Analysis

Combining the SM and soil texture, we created six classes (0: no conditions met, 1: sand or clay threshold met, 2: sand and clay met, 3: only SM threshold met, 4: one soil type and SM met, 5: all conditions met). Figure 2 shows a subset of the threshold maps generated and how the area of each of these conditions changed over time. Table 1 summarizes the breakdown of classifications for the locust observations 9 weeks prior. The location of the DL observations matched the overall optimal conditions (class 5) 9 weeks prior for 62.5% of the time. It can be seen that, in important breeding periods, the majority of the data were captured in the most optimal class, >80% of the time in the early breeding cycle (October–November), and >60% in the latter cycle (March–April) on average.

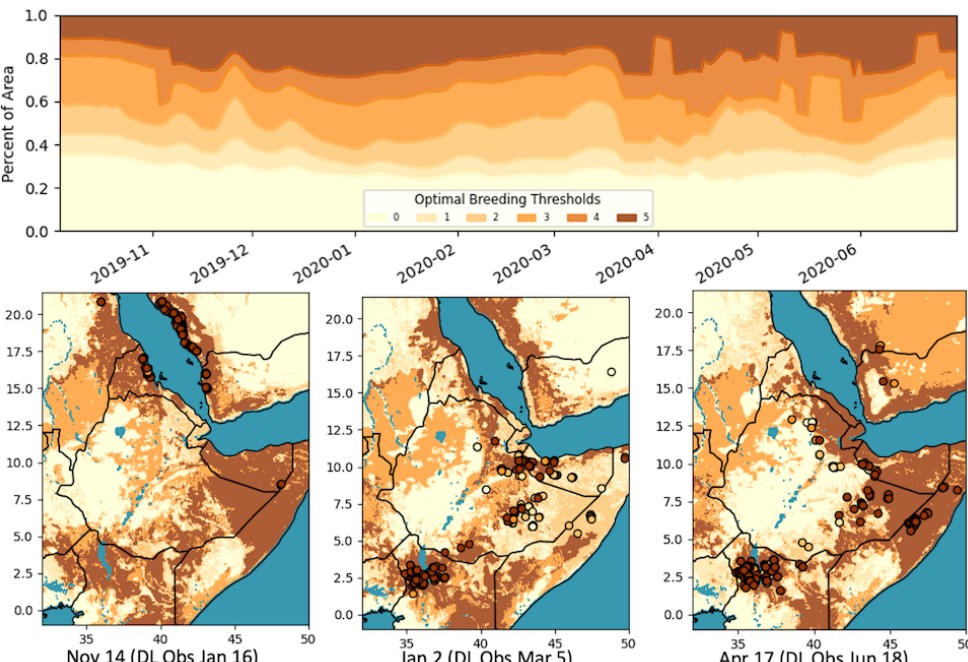

**Figure 2.** Optimal DL breeding threshold maps and the percentages of areas under each classification over time. Breeding thresholds: 0: no conditions met, 1: sand or clay threshold met, 2: sand and clay met, 3: only soil moisture (SM) threshold met, 4: one soil type and SM met, 5: all conditions met.

When considering other classes, such as 2 and 4, where only the soil type or SM was met (less optimal, but still triggering some conditions sufficient for breeding), the percentage of data captured was greatly increased (87.1% on average), especially in the latter cycle, though that would also contribute to a less specific (greater area) classification

(i.e., false positives). Additionally, whereas the soil texture and moisture datasets by themselves had classification areas of 30%, and 51%, respectively, the combined class 5 was, on average, significantly less, at 20% (17% during the main breeding cycles). The classification map identified where breeding is most likely to occur (class 5), areas in which some conditions are met and breeding may occur (classes 2–4), and areas where breeding is not expected. As we recognize the errors associated with both the soil type and SM datasets, the classification map is essentially providing a level of uncertainty in the likelihood of breeding conditions.

**Table 1.** Percentages (%) of DL observations under each threshold analyzed by month at the level of one pixel (3 × 3 km). Brackets show the corresponding percentages for a buffer of 3 × 3 pixels (9 × 9 km). See the text or Figure 2 for threshold definitions.

| Month | Breeding Threshold | | | | | |
|---|---|---|---|---|---|---|
| | **0** | **1** | **2** | **3** | **4** | **5** |
| 10 (*n* = 430) | 2.0 [0.0] | 2.1 [0.0] | 5.6 [5.3] | 0.9 [0.0] | 7.1 [0.9] | 84.0 [93.7] |
| 11 (*n* = 1003) | 2.0 [0.1] | 0.5 [0.1] | 4.9 [1.1] | 2.9 [0.0] | 8.1 [0.5] | 83.4 [98.2] |
| 12 (*n* = 339) | 2.9 [1.2] | 7.3 [3.5] | 15.2 [15.0] | 9.1 [0.0] | 18.5 [6.2] | 46.9 [74.2] |
| 1 (*n* = 351) | 3.0 [1.2] | 4.4 [0.8] | 16.6 [14.8] | 20.2 [6.2] | 22.3 [5.1] | 33.5 [71.7] |
| 2 (*n* = 224) | 19.6 [0.8] | 8.2 [3.3] | 26.9 [26.8] | 11.4 [4.1] | 14.7 [8.1] | 19.2 [56.9] |
| 3 (*n* = 1051) | 3.6 [0.7] | 2.9 [0.6] | 33.6 [3.0] | 7.4 [2.5] | 7.1 [2.6] | 45.3 [63.5] |
| 4 (*n* = 2763) | 0.9 [0.2] | 2.0 [0.2] | 9.2 [4.8] | 9.9 [0.1] | 12.6 [3.5] | 65.4 [90.1] |
| Average (wt) | 2.1 [0.4] | 2.5 [0.5] | 13.2 [9.7] | 0.083 [1.1] | 11.4 [3.5] | 62.5 [85.2] |

To better understand the spatial distribution of classes and help discuss the limitations, we highlighted a collection of DL observations in May. Figure 3 illustrates the classification map for 14 March, which corresponds to 265 DL observations between 14 and 18 May. In this time period, most locust observations corresponded to a classification of 5 or 2, in line with the broader results. In this case, class 5 represents 57% of the data, while class 2 (optimal soil, outside SM threshold) represents 31%.

This example was chosen because it showcases the sensitivity and the correlation of classes 2 and 5. When looking at the locust observations around the Shebelle river in south-eastern Ethiopia (b) or the areas near the northern Somali–Ethiopia border (c), just south of Djibouti, the SM-driven class 5 shows a distinct signal in each of these locations; however, the DL observation may exist one or two pixels outside class 5. This could be due to the fact that the locust had moved beyond the 3 km pixels, spatial/temporal errors present in the SM model, or the fact that the nine-week prior assumption was not accurate. This is an inherent error in the analysis; if the DL observed were less mature, the date of estimated egg laying will be off. This error is somewhat mitigated when averaging the SM at the weekly scale.

To better characterize the impact of this error, we can apply a buffer to the observations to see how often a point is missed by the difference (spatially) of one pixel. A 3 × 3 pixel buffer was applied (9 × 9 km) for each observation, as described above. The results of the buffer analysis are noted in brackets in Table 1. It can be seen that the results dramatically improved, indicating the extent to which the small spatial errors can impact the data. On average, the buffer improved the overall results from 62.5% to 85.2%. The biggest improvement was during the "off-cycle" observations (December, January, February), improving from 35% to 69% (wt. average) in the data under the optimal threshold. The off-cycle months were during the dry period between the two rainy seasons, and precipitation was limited and more sporadic, making it harder to resolve the spatial extent precisely, thus exacerbating the type of spatial errors discussed in Figure 3. Perhaps a larger source of uncertainty is the assumption of the DL maturity. As can be seen in Figure 3b, if the estimated breeding period was assumed to be 1–2 weeks later, the SM threshold would have been met. Given the DL observation data that we have, there is no certainty of

information on the age of the hoppers. We know that the majority are in their mid to late instar; however, the lengths of these individual stages vary widely and are an uncertainty that we must accommodate.

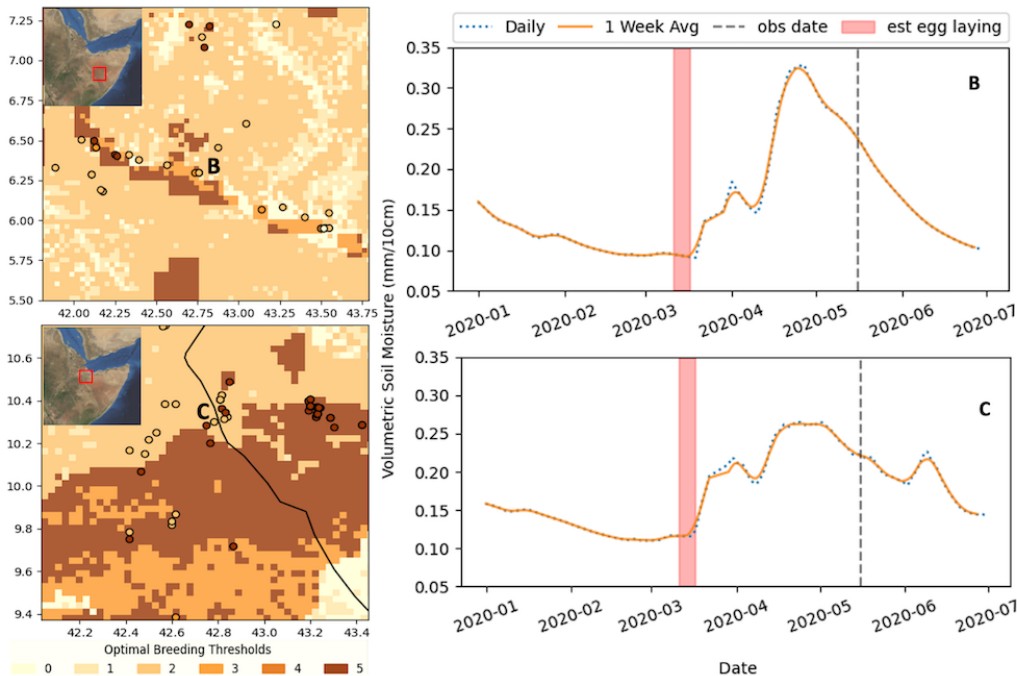

**Figure 3.** Optimal breeding classification map for 14 March (weekly average SM), which corresponds to 265 DL observations between 14 and 18 May, and the corresponding SM time series for two example cases in March for two regions in Ethiopia: B—Shabelle River and C—just south of Djibouti. The approximate location of the time series is annotated in the map in red. See the text or Figure 2 for threshold definitions.

### 3.4. Discussion

The improvement as a result of buffering can be explained, in part, by the inherent errors in the datasets, models, and assumptions in this analysis. Whereas the confidence is highest in the soil texture dataset (especially since they were aggregated up), the timing of the SM combined with the assumption of DL maturity created a lot of uncertainty in space and time. The error due to this uncertainty can be as large as 30% or more, as the buffer analysis suggests.

To attempt to better understand these uncertainties, we evaluated the sensitivity of the DL maturity, ranging from 5 to 9 weeks. While the nine-week average was best overall, most scenarios found that category 5 was met 60% of the time or more. The explanation for this can be seen in the time series plots of Figure 3. It turns out that the nine-week window occurred most often right at a major change, such as an increase in SM after a long recession. In most cases, the SM continued in the "optimal" range for several weeks. This indicates that nine weeks is roughly the earliest at which breeding conditions can be seen; however, it is likely that areas will stay optimal for several weeks. The association of the laying period with an increase in soil moisture is consistent with rainfall providing green vegetation for adult maturation and ample green vegetation for the survival of offspring nymphs [15–17].

Lastly, the amount of information added was quantified by analyzing the areas under which the thresholds occurred, with the understanding that the less area covered, the more accurate (fewer false positives) the "optimal breeding" dataset was. Indeed, though the areas classified as optimal were less than 20% of the study area, on average, this resulted in large areas of false positives. However, the purpose of this analysis was not to predict DL; instead, it was to evaluate the ability of soil texture and moisture conditions to be

used to better understand the breeding conditions in order to assist in monitoring efforts. The DL habit is expansive, and any deduction of areas that are likely have locust could be immensely helpful to survey efforts and for informing more efficient control measures. To accurately predict DL, more information, such as vegetation and temperature, is crucial. DL prediction models utilize many datasets, and DL experts are required to interpret those results. Here, we show that soil texture and land surface model SM datasets can be a useful component in that analysis. While this analysis focuses on historical DL observations, the LIS land surface model can be used to forecast and combine the native-resolution (250 m) soil texture data, potentially adding more value to the DL monitoring efforts.

## 4. Conclusions

In this study, we highlighted the ability of model-derived SM estimates to contribute towards broader locust monitoring activities. Model-estimated SM conditions can represent conditions beyond the surface layer, where DL egg-laying occurs (5–10 cm). There is a distinct signal that can be seen from the modeled SM and soil type in predicting the egg-laying areas of DL. When considering the most optimal conditions (all thresholds met), the soil texture combined with the modeled SM content predicted the estimated DL egg-laying period 62.5% of the time. Accounting for the data errors and uncertainties, a $3 \times 3$ pixel buffer increased this to 85.2%. By including the SM, the areas of optimal egg-laying conditions decreased from 33% to less than 20% on average (this can be interpreted as a >30% reduction in false positives). This paper shows that, though there are limitations to applying this analysis to DL monitoring efforts, there is a clear value in the added information that SM and texture provide in understating DL breeding conditions. This can assist in more targeted survey efforts, thus informing more efficient control measures. Additionally, by using a satellite-assisted modeled SM product, the barrier to operationalization and forecasting is reduced, thus potentially providing information products all along the locust management decision-making chain.

**Author Contributions:** Conceptualization, W.L.E., V.M., A.S.L., J.B.R. and K.C.; methodology, W.L.E., V.M., A.S.L. and K.C.; modeling, C.B.B. and J.L.C.; formal analysis, W.L.E.; writing—original draft preparation, W.L.E.; writing—review and editing, V.M., A.S.L., J.B.R. and K.C.; visualization, W.L.E. All authors have read and agreed to the published version of the manuscript.

**Funding:** Funding for this research was provided by National Aeronautics and Space Administration (NASA) through SERVIR. SERVIR is a joint program led by the US Agency for International Development (USAID) and NASA.

**Institutional Review Board Statement:** Not applicable.

**Informed Consent Statement:** Not applicable.

**Data Availability Statement:** The data are available publicly. Desert locust observations are available from the FAO (https://locust-hub-hqfao.hub.arcgis.com/ (accessed on 20 March 2021)) and the modeled soil moisture used in this study is available from SERVIR (https://gis1.servirglobal.net/data/locusts/east_africa/ (accessed on 20 March 2021)).

**Conflicts of Interest:** The authors declare no conflict of interest.

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
