# Peer review of "Detecting Desert Locust Breeding Grounds: A Satellite-Assisted Modeling Approach"

_remotesensing, doi:10.3390/rs13071276_

Round 1

Reviewer 1 Report

Dear authors, please address the followings

  1. Desert Locust (DL, Schistocraca gregaria), please provide a picture so that reader can understand the kind of species. Because, my thoughts on this go-everywhere e.g. a reptile, insect, etc.
  2. Enlarge, Fig.2 and Figure 3.
  3. In the method section, please add a table documenting datasets used, resolution, source, and application in the context of the present paper. Following the corresponding explanation in each section and sub-section. Furthermore, processing workflow could be helpful for readers the reproduce similar studies.

Author Response

Reviewer #1

Desert Locust (DL, Schistocraca gregaria), please provide a picture so that reader can understand the kind of species. Because, my thoughts on this go-everywhere e.g. a reptile, insect, etc.

We thank the referee for their valued review. Given the figure and length limitation of the MDPI communication format, we feel that adding a picture would require a trade off of existing information. It’s our hope that if a reader has further questions on DL, basic information (including pictures) is readily available online and the materials referenced.

Enlarge, Fig.2 and Figure 3.

We agree that the figures are of lesser quality in the draft version of manuscript. Although we have provided high resolution images with the submission and believe that the final version will reflect the higher quality images. 

In the method section, please add a table documenting datasets used, resolution, source, and application in the context of the present paper. Following the corresponding explanation in each section and sub-section. Furthermore, processing workflow could be helpful for readers the reproduce similar studies.

We thank the reviewer for these suggestions. Since there are only three datasets used, we feel a table describing them would be redundant and we are limited in space in a communications article. As you mentioned, each of the datasets are described in detail (sources, resolution, etc.) in the respective subsections. We do appreciate the need to further detail our workflow and have added several sentences detailing such, per your suggestion: Lines 84-102

To identify the optimal breeding conditions we used known locust observation from FAO, looked back to the estimated breeding period, and considered the soil conditions that were present. First, the 10th and 90th percentile values of soil texture (clay and sand) were determined and the percent area coverage was evaluated for each. Next, the DL development cycle was used to determine an approximate egg laying time frame. Soil moisture was aggregated to the weekly time scale to determine the estimated conditions during egg laying and the range of values and area of coverage was evaluated and compared against current literature. The two attributes were then combined to create a range of breeding condition thresholds and evaluated against the known DL observations. To better understand the temporal and spatial uncertainty, a 3x3 (9km x 9km) buffer was applied to the final threshold and discussed. With the lack of “no locust” observations, the results are evaluated in terms of classified areas and discussed in context of current monitoring efforts and existing literature.

Reviewer 2 Report

The manuscript entitled "Detecting desert locust breeding grounds: a satellite assisted modeling approach" was quite well-written but needs some adjustments. The authors presented an interesting approach using remote sensing data to monitor desert locust and so I consider it relevant to the Remote Sensing journal. However, this manuscript needs a proper revision before publishing as follows:

Specific comments

Abstract

Line 6 - Please, present both the spelled-out and short form the first time you use an abbreviation revising throughout the manuscript. 

Introduction

Line 15 - Scientific names of species have to be italicised.

Line 17-18 - You do not need to explain you will use certain abbreviation here onwards. Please, just refer to the abbreviation (e.g. DL).

Line 26-27 - Please, move the aim of your study to the last paragraph with the hypothesis. At the introduction section, the context and need work together as a "funnel". 

Line 47-48 - It can be one paragraph. 

Line 50-51 - Did they evaluate both 40-km and 1-km raster resolution?

Data and Methods

Line 90-91 - This sentence should be removed or moved to introduction (hypothesis and aims)

Line 84-97 - Do the authors really think it is necessary an introduction in this section? Line 94, for example, is a copy and write of some parts presented in the Abstract (e.g. Line 7). Thus, my suggestion is to remove it or move it to other subsections of M&M.

Figure 1. Please, improve this low resolution image replacing for a 300 dpi one. Other aspects:  (1) what's the CRS? Please, add it to the fig.; (2) add a legend for the points; (3) change the legend in fig.1b because it leads to misunderstandings. You wrote that circle stands for individual and square for bands. However, there are two squares. Moreover, change the color inside the label for hoppers and bands. It demands time to understand the entire figure. I think if you follow this recommendation, the fig. will print a quick summary of all info.

Lines 123-133 (Subsection 2.2) - Is that a result/discussion instead of method? 

Line 133 - [...] (in line [...]). If it's a result/discussion, please add the appropriate citation. If not, please remove it. 

Line 135 and 154 - The subsections 2.3.1 and 2.3.2 can be joint. No need for writing down "NOAH Soil Moisture from LIS and ISRIC Soil Classification". By the way, the authors are using the soil texture not the soil classification.

Results and Discussion

1. Please, add an uncertain map of the classification.

Line 259-260 - This description should be in the methods. I strong recommend the authors to add a section or paragraph explaining how they deal with different raster resolution in the methods (2.3).

Major comments:

1. How did you normalise/fit the two different rasters?

2. Is there any issue when downscaling? Is it better to uppscale or downscale?

3. The analysis would be more efficient if the soil texture was uppscaled to 3-km. 

4. Please, improve all low resolution images. I would like to see one more figure that highlights specific areas for the spatial distribution of classes. The authors can zoom in. It will add valuable visual content for the discussion and readers to see the patterns associated with DL.

Author Response

Reviewer #2

The manuscript entitled "Detecting desert locust breeding grounds: a satellite assisted modeling approach" was quite well-written but needs some adjustments. The authors presented an interesting approach using remote sensing data to monitor desert locust and so I consider it relevant to the Remote Sensing journal. However, this manuscript needs a proper revision before publishing as follows:

We thank the referee for a thorough review and have corrected and/or responded to each point you highlight.

Specific comments

Abstract

Line 6 - Please, present both the spelled-out and short form the first time you use an abbreviation revising throughout the manuscript. 

Corrected and thanks for this reminder we kept this in mind throughout the manuscript revision.  

Introduction

Line 15 - Scientific names of species have to be italicised.

Corrected 

Line 17-18 - You do not need to explain you will use certain abbreviation here onwards. Please, just refer to the abbreviation (e.g. DL).

Done

Line 26-27 - Please, move the aim of your study to the last paragraph with the hypothesis. At the introduction section, the context and need work together as a "funnel". 

Thanks for this comment and we understand your reasoning, however, given the ‘communications’ nature of this paper, we prefer to state the objective up front to ensure the reader can have the context needed when reading the introduction and background. 

Line 47-48 - It can be one paragraph. 

Corrected

Line 50-51 - Did they evaluate both 40-km and 1-km raster resolution?

The paper referenced evaluated the 1-km downscaled product 

Data and Methods

Line 90-91 - This sentence should be removed or moved to introduction (hypothesis and aims)

We removed this sentence

Line 84-97 - Do the authors really think it is necessary an introduction in this section? Line 94, for example, is a copy and write of some parts presented in the Abstract (e.g. Line 7). Thus, my suggestion is to remove it or move it to other subsections of M&M.

We thank the reviewer for their suggestion, however, we believe that this intro provides a good summary of the methodology before talking in depth about each component. We have added a more detailed description of the methodology in this section. Lines 84-102

Figure 1. Please, improve this low resolution image replacing for a 300 dpi one. Other aspects:  (1) what's the CRS? Please, add it to the fig.; (2) add a legend for the points; (3) change the legend in fig.1b because it leads to misunderstandings. You wrote that circle stands for individual and square for bands. However, there are two squares. Moreover, change the color inside the label for hoppers and bands. It demands time to understand the entire figure. I think if you follow this recommendation, the fig. will print a quick summary of all info.

We thank the reviewer for the suggestions to make this figure more clear. We agree, the goal of this figure is to provide a clear summary. There are only squares and circles in the image, hopefully the high res figure will make this more clear. However, there are more than 6000 points and the objective of the figure is not to identify each but to see the trends in the observation dates associated with fig1b. We took the extra time to ensure the colors in fig1b match that of figure 1a so as to not need another legend as explained in the caption. We took your advice and replaced the legend color in fig1b. Hopefully, these improvements will allow for a clearer summary. We added the CRS (WGS84) to the caption.

Sometimes the images are produced at lower quality in the draft. Nevertheless, we have uploaded a higher resolution image and believe that the final version will reflect the higher quality images.

Lines 123-133 (Subsection 2.2) - Is that a result/discussion instead of method? Line 133 - [...] (in line [...]). If it's a result/discussion, please add the appropriate citation. If not, please remove it. 

Thanks for highlighting this oversight. We have added the WMO/FAO reference.

Line 135 and 154 - The subsections 2.3.1 and 2.3.2 can be joint. No need for writing down "NOAH Soil Moisture from LIS and ISRIC Soil Classification". By the way, the authors are using the soil texture not the soil classification.

Thank you for this suggestion, since they are different datasets, we think it makes sense to  keep them separate, however, as you correctly point out, we are using soil texture, not classification, and corrected it as such.  

Results and Discussion

  1. Please, add an uncertain map of the classification.

We are unsure what exactly the reviewer is referring to. We have addressed the classification uncertainty in the buffer analysis pointing to uncertainties of the inputs and assumption.. We feel this adequately characterizes the uncertainties of the methodology employed. As we mentioned in the text, the threshold map itself acts somewhat as a map of uncertainty where classes 2 and 4 show some condition optimal for breeding and class 5 shows a strong likelihood of optimal breeding conditions.

Line 259-260 - This description should be in the methods. I strong recommend the authors to add a section or paragraph explaining how they deal with different raster resolution in the methods (2.3).

We thank the reviewer for this comment and have added the aggregation step in the methods section. Line 170

Major comments:

  1. How did you normalise/fit the two different rasters?

Both datasets are published in the World Geodetic System 1984 (WGS84) reference system (EPSG:4326). To “fit” the two rasters, we aggregated the ISRIC soil texture to 3km using nearest neighbor. 

  1. Is there any issue when downscaling? Is it better to uppscale or downscale?

We did not perform any downscaling. When aggregating the soil texture dataset to 3km to match that of the LIS soil moisture output, we used the nearest neighbor approach  as reported in section 3.1

  1. The analysis would be more efficient if the soil texture was uppscaled to 3-km. 

We agree and this was the approach used in the study. We added a sentence in section 2.3.2 to make this more clear. 

  1. Please, improve all low resolution images. I would like to see one more figure that highlights specific areas for the spatial distribution of classes. The authors can zoom in. It will add valuable visual content for the discussion and readers to see the patterns associated with DL.

As mentioned above, we have included a high resolution image for each figure. 

We agree with your suggestion to zoom in to show the specific spatial distribution and patterns. Instead of adding another figure, we have edited figure 3 to show individual zoomed-in areas.

Thanks again for all your comments and suggestions, we feel your input has improved the manuscript. 

Reviewer 3 Report

Dear authors, thanks for your paper, it is very interesting and relevant. I see there only one missing aspect and it is clime. This is indirectly mentioned in soil moisture and also in analysis of single months, but I think, that for better understanding of problem, it will be useful to mentioned it explicitly in text.

Author Response

Reviewer #3

Dear authors, thanks for your paper, it is very interesting and relevant. I see there only one missing aspect and it is clime. This is indirectly mentioned in soil moisture and also in analysis of single months, but I think, that for better understanding of problem, it will be useful to mentioned it explicitly in text.

We thank the referee for their review and complement. Per your suggestion, we have added a few sentences to the study area section. Lines 105-110

The climate of East Africa ranges from hot arid to tropical savanna [beck] as a result of the complex terrain and in general, is anomalously dry for equatorial regions [camberlin]. Rainfall across the region can be characterized by a bimodal regime with the movement of the Intertropical Convergence Zone resulting in the “long rains” from March - May and the “short rains'' from October - December [camberlin].

Author Response

Reviewer #4

The results of this study are a good proof of concept for adding these types of satellite estimations to the FAO locust monitoring and forecasting system. The desert locust area is huge and any method that could identify areas more likely to have locusts would be useful. Therefore, in practical terms, the use of this type of assessment could be in monitoring, more or less in real time, soil moisture in sandy habitats as a place to concentrate more survey effort. Not that all effort should be there because of shortcomings in data and variance in behaviour of locusts, but more effort in areas identified as more likely as part of risk management. The above points re usefulness in concentrating survey efforts should be included in the discussion.

We thank the reviewer for a thoughtful and thorough review. Your point about the concentration of survey efforts is an important consideration and we have included this point in both our discussion and conclusion. Lines 311-313 and 332-334 

Line 108: “…gregarious “bands” that march …..”

Corrected 

Lines 124-126: incubation period 10-30 days at the soil temperatures: why 10 days+ when Figure 1 has 14 days+? And why 4-6 weeks for hoppers in text yet Figure 1 has 35-45 days (5-6.4 weeks)?

Thank you for noticing the inconsistency. The periods in Fig 1 are correct, and we edited the text to match.

Line 136: ..using the NOAA land surface…

“Noah”, is correct. We have added a specific reference to help clarify for other readers. 

Ek, M. B., K. E. Mitchell, Y. Lin, E. Rogers, P. Grunmann, V. Koren, G. Gayno, and J. D. Tarpley, 2003: Implementation of Noah land surface model advances in the National Centers for Environmental Prediction operational mesoscale Eta model. J. Geophys. Res., 108 (D22), 8851, doi:10.1029/2002JD003296.

Line 138: Change “( 3-km) lat/lon” to “(3-km) lat/long”

There was supposed to be a ~ there that was translated in the latex editor - corrected and added long vs lon 

Line 158: “…texture at depths from the surface…” add “the”

corrected

Line 177: “..operational DL monitoring…” Also No space between “..useful).The”

fixed

Line 193: No space between “…prior.The…”

fixed

Lines 195-198: Why italics?

Initially, we thought this description was more of an aside, however, after further thought, we agree and have kept the text without italics

Line 197: ( 20%) should be (20%)

There was supposed to be a ~ there that was translated in the latex editor - corrected. 

Line 201: “...previous studies like those of ….. that show a mean…”

Corrected 

Line 207: With the top layer of soil… You have two tops!

Corrected, thanks. 

For Table 1, explain more clearly what the two numbers mean—something like:

“Table 1. Percentage of DL observations under each threshold, by month analysed at the level of one pixel (3km x 3 km). Brackets show the corresponding percentages for a buffer of 3x3 pixels (9km x 9 km). See text or Figure 2…….”

Thank you for this suggestion!, we have taken your advice and tried to be more descriptive in the table title. 

Numbers in Table 1 vs numbers in text. In some places, it is difficult to find what you are referring to in the text. Line 226: 62.5% is the same as the average 0.625 average under 5 in Table 1. But it is not clear what data you are referring to in Table 1 for the improvements when you use 3x3 pixel

This is a great observation. To help with the overall clarity of the results, we have changed the table to report in percentages (%) as the text does. It also makes the table more readable. We have also edited the lines in question to read as such: Line 239

The location of the DL observations matched the overall optimal conditions (class 5)  9 weeks prior 62.5% of the time. It can be seen that in important breeding periods the majority of the data is captured in the most optimal class,> 80% of the time in the early breeding cycle (Oct-Nov), and > 60% in the latter cycle (March-April), on average. 

This information is still coming from the same column, just highlighting the important breeding periods of Oct-Nov and Mar-Apr. We hope this is more clear now. 

Line 262: “improved the overall results from 68% to 90%” but Table 1 has 0.625 increasing to 0.852.

This was a typo and has been corrected. 

Line 263: “the ‘off cycle’ observations (from 38% to 76%)” Explain which months are off cycle that get 38% increasing to 76% as if I include months 1, 2, and 3 and use data under 5, I get (0.335+0.192+0.453)/3=0.327 or 33%.

Thank you for your thorough review. You are correct. The text must have been reporting earlier, preliminary results and apologize for the confusion and inconsistency. Table 1 values are correct and the percentages are weighted averages. Edited the text as follows: Line 274

The biggest improvement was during the “off cycle” observations (Dec,Jan,Feb), improving from 35 to 69% (wt average) under the optimal threshold. 

Similarly Lines 308-309 in Conclusions: the 62% is close to the 62.5% of line 226, but where does the 90.2% come from: is that the 90% in line 282? Also be consistent: 62.5%, 90.2%.

Corrected. 

Figure 3. Since not everyone is familiar with the geography of Ethiopia, I suggest putting arrows on the map to show where B and C are on the map and say in Figure 3: “....between May 14-18 and corresponding SM time series in March for two regions in Ethiopia: B-Shabelle River and C-just south of Djibouti…”

Thanks for this suggestion. Originally we did just that, annotating the B and C in the figure. We will make sure it is there and will edit the caption as you suggest. 

Line 252: “…due to the fact that the locusts had moved…”

Line 254: “..if the DL observed were less mature”

corrected 

Line 252-256: In spite of these various sources of error in estimating the laying date, using the 8-10 weeks (mean 9 weeks) before nymphs/bands is about the time of increased soil moisture. As you say in lines 267-269, if the estimated breeding period was assumed to be 1-2 weeks later, the SM threshold would have been met. Perhaps you should clarify why there is this correlation with soil moisture by including in the discussion something like: “The association of laying period with an increase in soil moisture is consistent with rainfall providing green vegetation for adult maturation and laying (references), soil moisture for egg survival and development (references) and ample green vegetation for survival of offspring nymphs (references).

This is a great point and we agree this will add to the discussion. We have included such points in the discussion around Line 300.

Thanks again for all your comments and suggestions, we feel your input has improved the manuscript. 

Round 2

Reviewer 3 Report

Dear authors, thanks, good paper